# Effects of Environmental Hypoxia on Serum Hematological and Biochemical Parameters, Hypoxia-Inducible Factor (*hif*) Gene Expression and HIF Pathway in Hybrid Sturgeon (*Acipenser schrenckii* ♂ × *Acipenser baerii* ♀)

**DOI:** 10.3390/genes15060743

**Published:** 2024-06-05

**Authors:** Yuanyuan Ren, Yuan Tian, Bo Cheng, Yang Liu, Huanhuan Yu

**Affiliations:** 1Key Laboratory of Control of Quality and Safety for Aquatic Products, Ministry of Agriculture and Rural Affairs, Chinese Academy of Fishery Sciences, Beijing 100141, China; renyuany@cafs.ac.cn (Y.R.); chengb@cafs.ac.cn (B.C.); 2Key Laboratory of Mariculture (Ocean University of China), Ministry of Education, Qingdao 266003, China; tianyuan@ouc.edu.cn; 3State Key Laboratory of Mariculture Biobreeding and Sustainable Goods, Yellow Sea Fisheries Research Institute, Chinese Academy of Fishery Sciences, Qingdao 266071, China; yangliu@ysfri.ac.cn; 4Fisheries Science Institute, Beijing Academy of Agriculture and Forestry Sciences, Beijing 100068, China

**Keywords:** hypoxia, physiological response, hypoxia-induced factor, HIF pathway, hybrid sturgeon

## Abstract

Hypoxia is a globally pressing environmental problem in aquatic ecosystems. In the present study, a comprehensive analysis was performed to evaluate the effects of hypoxia on physiological responses (hematology, cortisol, biochemistry, *hif* gene expression and the HIF pathway) of hybrid sturgeons (*Acipenser schrenckii* ♂ × *Acipenser baerii* ♀). A total of 180 hybrid sturgeon adults were exposed to dissolved oxygen (DO) levels of 7.00 ± 0.2 mg/L (control, N), 3.5 ± 0.2 mg/L (moderate hypoxia, MH) or 1.00 ± 0.1 mg/L (severe hypoxia, SH) and were sampled at 1 h, 6 h and 24 h after hypoxia. The results showed that the red blood cell (RBC) counts and the hemoglobin (HGB) concentration were significantly increased 6 h and 24 h after hypoxia in the SH group. The serum cortisol concentrations gradually increased with the decrease in the DO levels. Moreover, several serum biochemical parameters (AST, AKP, HBDB, LDH, GLU, TP and T-Bil) were significantly altered at 24 h in the SH group. The HIFs are transcription activators that function as master regulators in hypoxia. In this study, a complete set of six *hif* genes were identified and characterized in hybrid sturgeon for the first time. After hypoxia, five out of six sturgeon *hif* genes were significantly differentially expressed in gills, especially *hif-1α* and *hif-3α,* with more than 20-fold changes, suggesting their important roles in adaptation to hypoxia in hybrid sturgeon. A meta-analysis indicated that the HIF pathway, a major pathway for adaptation to hypoxic environments, was activated in the liver of the hybrid sturgeon 24 h after the hypoxia challenge. Our study demonstrated that hypoxia, particularly severe hypoxia (1.00 ± 0.1 mg/L), could cause considerable stress for the hybrid sturgeon. These results shed light on their adaptive mechanisms and potential biomarkers for hypoxia tolerance, aiding in aquaculture and conservation efforts.

## 1. Introduction

Sturgeons are notable ancient fish species with unique properties, such as low evolution rates, characteristics of the Chondrosteus species, and some other features between Chondrichthyes and Osteichthyes [1,2], which have attracted considerable scientific attention. In addition, they are economically important species, popularly produced in aquaculture. At present, sturgeon culture is growing worldwide, and China has become the largest producer of sturgeons since 2000 [3]. In China, the hybrid sturgeon (*A. schrenckii* ♂ × *A. baerii* ♀) is the dominant cultured species due to its rapid growth rate and strong resistance to disease [4,5,6].

Oxygen is one of the most important environmental factors for the survival, growth, development and reproduction of aquatic animals. However, hypoxia often occurs in coastal, estuarine and aquaculture environments, arising from a single factor or the combined action of several factors, such as the tidal cycle, eutrophication, global warming, freshwater runoff and stratification of the water column [7,8,9,10]. In recent years, hypoxia has occurred more frequently due to human activities [11,12]. As a result, it has caused great economic losses and hindered the development of aquaculture [13,14]. To design effective strategies to reduce the negative effects of hypoxia on aquaculture, it is of great significance to evaluate the effects of hypoxia on physiological responses. These related studies have been reasonably well documented in teleosts. However, limited attention has been paid to sturgeons, particularly the hybrid sturgeon.

Therefore, a comprehensive analysis was performed to evaluate the effects of hypoxia stress on several well-known biomarkers in hematology and biochemistry, as well as the *hif* gene expression and HIF pathway of the hybrid sturgeon. The changes in hematological indexes, such as the red blood cell (RBC) count and the hemoglobin (HGB) concentration, could directly reflect the oxygen uptake and carrying capacity [15,16,17]. In addition, the increase in cortisol hormone concentration, synthesized and released by the interrenal cells of the head kidney, has been widely employed as a quantitative measure of stress in fishes [18] such as rainbow trout (*Oncorhynchus mykiss*) [19] and Amur sturgeon (*A. schrenckii*) [20]. Cortisol is able to induce related secondary responses to regulate important physiological processes and maintain homeostasis [21]. Biochemical characteristics are thought to be associated with the metabolism processes of aquatic animals. For example, the activities of aminotransferase (ALT) and aspartate aminotransferase (AST) are considered as the biomarkers to assess liver function, and elevated ALT and AST usually indicate impaired liver function. Glucose (GLU) and lactate concentrations (LA) are closely related to energy metabolism and ATP production [22,23]. Ni et al. [20] reported that the serum GLU of the Amur sturgeon first increased, then decreased during continuous hypoxia for 6 h. The inorganic ions concentrations could be used to detect the normal cell metabolism or cellular damage caused by hypoxia [24].

Hypoxia-inducible factors (HIFs) are transcription activators that function as master regulators, which would activate the expressions of more than 150 genes encoding proteins that regulate cell metabolism, survival, motility, basement membrane integrity, angiogenesis, hematopoiesis and other functions in response to hypoxia [25,26,27]. Briefly, HIFs are heterodimers composed of two subunits, HIF-α and HIF-β, which belong to the basic Helix-Loop-Helix-Per-ARNT-Sim (bHLH–PAS) superfamily [28,29]. To date, three paralogs of HIF-α (HIF-1α, HIF-2α, HIF-3α) and HIF-β (HIF-1β, HIF-2β and HIF-3β) have been identified and systematically characterized in numerous teleosts. By contrast, studies about the *hif* genes in sturgeons remain limited, and only *hif-α* were previously reported in Siberian sturgeon (*A. baerii*) [30], Persian sturgeon (*A. persicus*) [31] and white sturgeon (*A. transmontanus*) [32]. Recently, the HIF pathway, mainly composed of *hif* genes, has received much attention for its diverse biological functions [33,34]. Increasing studies suggest that the HIF pathway is present in virtually every cell of the body and plays important roles in orchestrating a whole cascade of downstream genes so as to allow organisms to acclimate to hypoxia environments [35].

In the present study, a complete set of *hif* genes was identified, and their expression patterns were determined after hypoxia, combined with hematological indexes, cortisol and biochemical parameters to assess the effects of hypoxia on the physiological responses of hybrid sturgeons. In addition, the expression levels of the function genes in the HIF pathway were also determined after the hypoxia challenge using RNA-Seq datasets.

## 2. Materials and Methods

### 2.1. Ethics Statement

Ethics approval for this study was obtained from the Institutional Review Board at Ocean University of China (Permit Number: 20141201), and all participants provided written informed consent. This study did not involve endangered or protected species and the experiments were performed in accordance with the relevant guidelines.

### 2.2. Hypoxia Challenge Experiment and Sample Collection

A total of 180 hybrid sturgeon adults (81.6 ± 6.3 g) were randomly selected from a local farm in Linyi, Shandong Province, China and acclimatized for a week in a circular tank (radius = 5 m, height = 5 m). During the acclimation period, the water temperature (19.6 ± 1.3 °C), pH (7.5 ± 0.2) and the DO (7.0 ± 0.4 mg/L) remained stabilized. Fish were fed to apparent satiation (around 1.8% body weight per day) with commercial compound diets (Ningbo Tech-bank Co., Ltd., Ningbo, Zhejiang province, China) until 24 h prior to the start of the hypoxia experiment.

After acclimation, the fish individuals were randomly divided into a control group (N, 7.00 ± 0.2 mg/L), a moderate hypoxia treatment group (MH, 3.5 ± 0.2 mg/L) and a severe hypoxia treatment group (SH, 1.00 ± 0.1 mg/L) in triplicate tanks (water volume 300 L), at a density of 20 individuals per tank. When the experiment started, the water of the MH and SH groups was deoxygenated for 1 h by bubbling nitrogen gas in order to reduce the oxygen concentration from 7.00 ± 0.2 mg/L to the desired threshold. The DO levels of the MH and SH groups were maintained relatively constant by a mixture of air and nitrogen gas. The DO levels of all the groups were continuously monitored using ODEON Multy 8320 dissolved oxygen meters (Ponsel, Caudan, France) during the whole experiment. Three individuals per tank were anesthetized with MS-222, and gill, liver (24 h) and blood samples were collected at 0 h and 1 h, 6 h, 24 h after hypoxia treatment. Gill and liver samples were immediately transferred into RNase-free tubes, quickly frozen in liquid nitrogen, and stored at −80 °C until RNA extraction. Blood samples were collected from the caudal vein using 1 mL syringes and divided into two parts. One portion was mixed with ethylenediamine tetraacetic acid (2.52 mg/mL) at the volume ratio 2:1 for hematological analysis. The other portion was kept at −4 °C for 4 h and then centrifuged for 10 min at 1500× *g* to obtain serum for the measurement of hormone and biochemical parameters.

### 2.3. Measurement of Serum Hematological Indexes, Cortisol Concentrations, and Biochemical Parameters

The RBC counts (10^12^/L) and the HGB concentration (g/L) were measured by electronic resistance methods using an automatic blood analyzer B1800 (Mindray, Shenzhen, China).

The cortisol concentrations of the serum samples were assayed by radioimmunoassay (RIA), which was conducted using iodine [I125]–cortisol kits (Tianjin Jiuding Medical Biological Company, Tianjin, China) with a γ-radiation counter SN-695B (Shanghai institute of applied physics, Shanghai, China). All procedures were performed according to the manufacturer’s instructions.

For biochemical analysis, serum samples were used to determine the concentrations of ALT (U/L), AST (U/L), alkaline phosphatase (AKP, U/L), α-hydroxybutyrate dehydrogenase (HBDH, U/L) and lactate dehydrogenase (LDH, U/L), GLU(mmol/L), total protein (TP, g/L), albumin (ALB, g/L), total cholesterol (TC, mmol/L), triacylglycerol (TG, mmol/L), total bilirubin (T-Bil, umol/L), lactic acid (LA, mmol/L), calcium (Ca, mmol/L), magnesium (Mg, mmol/L) and phosphorus (P, mmol/L) with the respective commercial assay kits, according to their manufacturer’s instructions, using a BS180 Automated Biochemistry Analyzer (Mindray, Shenzhen, China).

### 2.4. Identification of hif Genes in Hybrid Sturgeon

Firstly, the trimmed high-quality reads from the RNA-Seq datasets of the hybrid sturgeon (NCBI BioProject: PRJNA347887) were assembled into unique transcripts using Trinity v2.11.0 software [36]. Then, TBLASTN was employed to search the assembly transcripts with e-values of 1 × 10^5^. The amino acid sequences of the *hif* genes in humans (*Homo sapiens*), zebrafish (*Danio rerio*) and sturgeons (*A. sturio*), downloaded from NCBI, were used as queries. The lengths of protein products were predicted by ORF Finder (https://www.ncbi.nlm.nih.gov/orffinder/, accessed on 12 September 2023) based on the nucleotide sequences of the *hif* genes of the hybrid sturgeon. The conserved domains of protein products were identified and predicted by Simple Modular Architecture Research Tool (SMART) (http://smart.embl.de/, accessed on 12 September 2023).

### 2.5. Phylogenetic Analysis of hif Genes

To confirm the annotation of the *hif* genes in the hybrid sturgeon and investigate the evolutionary relationships, phylogenetic analysis was conducted using the predicted *hif* amino acid sequences of the hybrid sturgeon and several selected species, downloaded from the NCBI and ENSEMBLE databases, including human, mouse (*Mus musculus*), zebrafish, channel catfish (*Ictalurus punctatus*) and large yellow croaker (*Pseudosciaena crocea*). The amino acid sequences were firstly aligned using MSUCLE software (v5.1) with the default parameters. A phylogenetic tree was built using MEGA 7.0 software based on the neighbor-joining (NJ) method and the Jones–Taylor–Thornton model with 1000 bootstrap replicates [37].

### 2.6. RNA Extraction and Quantitative Real-Time PCR (qPCR) Experiment

The total RNA of each sample was extracted using the traditional TRIzol reagent (Invitrogen, Carlsbad, CA, USA) according to the manufacturer’s protocol. Then, the RNA was digested with RNase-free DNase I (Takara, Otsu, Japan) to remove genomic DNA contamination. The RNA concentration and quality were evaluated by the ratio of OD260/280 using NanoDrop 2000 (Thermo Fisher Scientific, Waltham, MA, USA). The integrity was determined by 1% agarose gel electrophoresis. Equal amounts (500 ng per sample) of RNA from 3 replicated samples per tank were pooled as one sample to minimize the variation among individuals. Then, the pooled RNA was reverse transcribed into cDNA using the PrimeScript^TM^ RT reagent kit (Takara, Otsu, Japan) for the detection of the mRNA level of *hif* genes by qPCR (gill samples) or using SMARTer™ PCR cDNA Synthesis Kit (Clontech, Mountain View, CA, USA) for RNA-Seq (liver samples), following the manufacturer’s instructions.

All the gene-specific primers for qPCR were designed using Primer 5 software, based on the nucleotide sequences of the hybrid sturgeon *hif* genes identified in this study, and 18S rRNA was set as the internal reference gene (Appendix A). Prior to qPCR, the specificity of these primers was tested by dissociation curve analysis. qPCR amplification was carried out on the Applied Biosystems 7300 machines (Applied Biosystems, Foster, CA, USA) using SYBR Premix Ex TaqTM kit (Takara, Shiga, Japan) under the following conditions: 95 °C for 30 s and 40 cycles of 95 °C for 5 s, Tm °C for 30 s, followed by 95 °C for 15 s, 60 °C for 1 min and 95 °C for 15 s. The qPCR reaction volume consisted of 20 μL, including 2 μL of cDNA, 10 μL of SYBR premix Ex Taq, 0.4 μL of forward primers, 0.4 μL of reverse primers, 0.4 μL of ROX Reference Dye, and 6.8 μL of ddH_2_O. All qPCR experiments were performed in three biological replicates with three technical replicates. The relative gene expression levels were calculated by the 2^−ΔΔCt^ method.

### 2.7. Meta-Analysis of HIF Pathway

RNA-Seq was performed on an Illumina Hiseq platform (Illumina, San Diego, CA, USA) with liver samples from hybrid sturgeon treated with normoxia (N group) and severe hypoxia (SH group) for 24 h. The sequencing data are available through the NCBI database (https://www.ncbi.nlm.nih.gov/, accessed on 8 December 2016) with a BioProject ID: PRJNA356676. The Illumina-based RNA-Seq datasets were used for the meta-analysis of HIF pathway. The bioinformatic pipelines have been constructed in the previous studies of zebrafish [38], gilthead sea bream (*Sparus aurata*) [39] and flounder (*Paralichthys olivaceus*) [40]. The high-quality reads were mapped to the above-mentioned assembly transcripts using Hisat2 v2.2.1 software [41]. The mapped reads were counted and submitted to DESeq2 R package (v1.34.0) for the calculation of expression levels.

### 2.8. Statistical Analysis

Data were presented as mean ± standard error (SE). SPSS 21.0 software was employed for statistical analyses. A one-way analysis of variance (ANOVA), followed by Duncan’s multiple range tests, was used to analyze the experimental data. Differences were considered to be significant at *p* < 0.05.

## 3. Results

### 3.1. Effects of Hypoxia on Hematological Indexes and Serum Cortisol

The hematological analysis indicated that the RBC counts and the HGB concentrations were significantly (*p* < 0.05) influenced by the hypoxia challenge (Figure 1). The RBC counts in the SH group were significantly (*p* < 0.05) increased at 6 h and 24 h after hypoxia (Figure 1A). No significant (*p* > 0.05) difference in the RBC counts was detected between the N and MH groups during the whole experiment. The HGB concentration in the MH group was induced remarkable (*p* < 0.05) at 6 h after hypoxia, while a significantly (*p* < 0.05) increased HGB concentration in the SH group was observed at both 6 h and 24 h after the hypoxia challenge (Figure 1B). The serum cortisol concentrations in the MH and SH groups were significantly (*p* < 0.05) induced by hypoxia to varying degrees (Figure 1C). Significant (*p* < 0.05) rises in cortisol concentrations in the MH group were observed at 6 h and 24 h after hypoxia challenge, which were 3.9 ng/mL and 4.3 ng/mL, respectively. In addition, the cortisol concentrations in the SH group were sharply increased from 1 h (3.5 ng/mL) after the hypoxia challenge and exhibited higher levels (> 8 ng/mL) at 6 h and 24 h after the hypoxia challenge (*p* < 0.05).

### 3.2. Effects of Hypoxia on Serum Biochemical Parameters

Serum enzyme activities, metabolites contents and inorganic ion concentrations in hybrid sturgeon were detected at 24 h after the hypoxia challenge (Table 1). For serum enzymes, the AST activities were significantly (*p* < 0.05) increased in the SH group. A significant (*p* < 0.05) decrease in the AKP activities was also found in the SH group. The enzyme activities of HBDH and LDH in the SH group were increased (*p* < 0.05) in comparison with the N and MH groups after hypoxia. Different DO levels had no significant (*p* > 0.05) effects on the ALT activity.

After 24 h of hypoxia, the contents of GLU, T-Bil and TP in the hybrid sturgeon were significantly increased (*p* < 0.05) in the SH group compared to the N and MH groups (Table 1). The LA concentration showed an increasing tendency with increasing DO levels (N > MH > SH). No significant (*p* > 0.05) difference was observed in serum TG, TC and ALB contents.

There was no significant (*p* > 0.05) change in serum inorganic ions contents as the DO level decreased.

### 3.3. Characterization of hif Genes in Hybrid Sturgeon and Phylogenetic Analysis

A total of six *hif* genes were identified in the hybrid sturgeon, including *hif-1α*, *hif-2α*, *hif-3α*, *hif-1β*, *hif-2β* and *hif-3β*. The characteristics of the hybrid sturgeon *hif* genes are summarized in Table 2, and their cDNA sequences were submitted to the GenBank database. In detail, the lengths of coding sequences of the hybrid sturgeon *hif* genes ranged from 1590 to 2562 bp, and the predicted protein sizes ranged from 530 to 854 aa. Additionally, all *hif* genes in the hybrid sturgeon commonly contained the HLH, PAS and PAC domains. The HIF-1a_CTAD domain was discovered in both *hif-1α* and *hif-2α*, while the HIF-1 domain only existed in *hif-3α*.

The phylogenetic analysis was employed to confirm the annotation of the *hif* genes in the hybrid sturgeon and evaluate the evolution relationships of the *hif* genes. The results demonstrated that the *hif* genes were clearly classified into six clades, including *hif-1α*, *hif-2α*, *hif-3α*, *hif-1β*, *hif-2β* and *hif-3β* (Figure 2). As expected, the *hif* genes of the hybrid sturgeon were clustered with their counterparts in selected teleost species, which provide additional evidence to support the annotation of the *hif* genes in the hybrid sturgeon. In addition, the phylogenetic result of the *hif* genes could be considered a clue to their evolution.

### 3.4. Expression Analysis of hif Genes after Hypoxia Challenge

In general, five out of six *hif* genes, including *hif-1α*, *hif-2α*, *hif-3α*, *hif-1β* and *hif-2β* in the hybrid sturgeon were significantly differentially expressed in response to hypoxia stress (Figure 3A–F). The expression levels of *hif-1a* in the SH group were significantly (*p* < 0.05) increased, with nearly 18-folds at 1 h after hypoxia in comparison with the N group, and reached the peak at 6 h, with an approximate 32-fold increase (Figure 3A). A highly up-regulated (*p* < 0.05) expression level of *hif-1a* in the MH group appeared only at 6 h after hypoxia (Figure 3A). As shown in Figure 3B, *hif-2a* displayed similar expression patterns with *hif-1a.* The expression levels of *hif-3a* in the MH and SH groups increased about 7- and 21-fold at 6 h after hypoxia, respectively (*p* < 0.05), and returned to normal levels at 24 h (Figure 3C).

Significant increases (*p* < 0.05) in the *hif-1β* expression in both the SH and MH groups were detected at 6 h and 24 h after hypoxia (Figure 3D). Compared to the N group, high increases (*p* < 0.05) in *hif-2β* were detected in the SH group at 1 h and 6 h after hypoxia (Figure 3E). The *hif-3β* expression was not affected (*p* > 0.05) by the DO levels (Figure 3F).

### 3.5. Meta-Analysis of Gene Expression in HIF Pathway after Hypoxia Challenge

The schematic diagram of the HIF pathway is shown in Figure 4. Five out of six *hif* genes, except for *hif-3a*, displayed increased expressions 24 h after hypoxia challenge. The expression of some HIF-dependent genes was also increased 24 h after hypoxia, including *glucose transporter protein type 1* (*glut1*), *glycogen synthase 1* (*gys*), *lactate dehydrogenase A* (*ldha*), *glycine amidinotransferase* (*gatm*), *angiopoietin-related protein 3* (*angptl3*) and *vascular endothelial growth factor* (*vegf*). The expression of the *von hippel–lindau* (*vhl*) gene was repressed, while the *prolyl hydroxylase domain-containing protein* (*phd2*) was stimulated by hypoxia in the liver tissues of the hybrid sturgeons.

## 4. Discussion

The hematological analysis indicated that the RBC count and the HGB concentration were significantly increased by the hypoxia challenge in the sturgeon hybrids examined in the current study. Similar results were also observed in the Amur sturgeon [20], the Atlantic sturgeon (*A. oxyrhynchus*) [42] and the turbot (*Scophthalmus maximus*) [43]. RBC and HGB are important components of the fish blood, being responsible for delivering oxygen to various parts of the body [44,45]. An increased RBC count and HGB concentration, caused by the release of erythropoietin into the circulation system, could improve the oxygen-carrying capacity of the blood in response to hypoxia stress and prevent tissue damage [46,47]. Cortisol, also known as hydrocortisone, is a steroid hormone produced by the adrenal gland. It plays a crucial role in the body’s stress response and helps regulate the blood glucose level, blood pressure, and immune function. However, a chronically elevated cortisol level has negative effects on health, including an increased risk of obesity, diabetes, and cardiovascular disease [48,49]. Serum cortisol concentrations were significantly induced by hypoxia to varying degrees, indicating that the hybrid sturgeons were subjected to stress after hypoxia. Elevated cortisol in response to hypoxia was also found in the Amur sturgeon [20]. However, the ability of the hybrid sturgeon to tolerate hypoxia seems to be stronger than that of the Amur sturgeon because the increase in cortisol in the hybrid sturgeon was observed after 6 h of hypoxia (3.5 mg/L), while increased cortisol levels were observed in the Amur sturgeon after 0.5 h of hypoxia (5 mg/L) [20].

As the enzyme biomarkers for assessing liver function, the changes in AST activities in the SH group indicated the damage of the liver function was caused by severe hypoxia stress [50,51]. AKP is usually regarded as one of the important multifunctional enzymes involved in metabolism and innate immune system functions [52]. The decreased AKP activities in the SH group revealed that severe hypoxia could cause an adverse impact on the metabolism and innate immune system of the hybrid sturgeon. HBDH and LDH, which are released into the peripheral blood when myocardial damage occurs, are markers of myocardial injury. The enzyme activities of HBDH and LDH in the SH group were increased after hypoxia, suggesting a negative effect of hypoxia on myocardial function [53]. In fishes, the energy demand was usually increased in response to environmental stressors [54]. GLU is a source of fuel for anaerobic metabolism, and LDH is necessary to catalyze the conversion of pyruvic acid to LA in glycolysis under anaerobic conditions [55,56]. The increased GLU, LDH activity and LA in the SH group suggested that anaerobic metabolism could make contributions to energy production in the hybrid sturgeon during hypoxia. However, Naya-Català et al. [39] highlighted a higher contribution of aerobic metabolism to the whole energy supply under hypoxia. The different results might be due to varied duration of hypoxia or different fish species. Previous studies have demonstrated that T-Bil is the major product of heme catabolism and provides important protection against oxidative stress-mediated diseases [57]. In our results, the elevated T-Bil in the SH group might be a response to prevent the damage caused by excessive reactive oxygen species under a hypoxia environment in hybrid sturgeons. Similar results were previously reported in goldfish (*Carassius auratus*) [58], in which the T-Bil levels were significantly increased in response to hypoxia. The TP content is usually used as one of the general indicators of the liver function in vertebrates, and either an increase or decrease in serum TP may indicate the occurrence of liver disease [59]. In the present study, serum TP was found to be significantly elevated in the SH group. Taken together with the significantly increased AST activity in the SH group, it was inferred that severe hypoxia for 24 h would cause hepatocyte damage.

Gills are considered the critical organs responsible for respiratory gas exchange and sensing in fishes [60,61,62]. Hence, the expression patterns of all six *hif* genes were examined in the gills of the hybrid sturgeons at 0 h, 1 h, 6 h and 24 h after hypoxia. In general, five of six *hif* genes, including *hif-1α*, *hif-2α*, *hif-3α*, *hif-1β* and *hif-2β,* in the hybrid sturgeons were significantly differentially expressed in response to hypoxia stress. As the master regulator of oxygen homeostasis, *hif-1α* is primarily responsible for the activation of numerous hypoxia-responsive genes to increase the O_2_ uptake and decrease the O_2_ demand under hypoxia [63,64]. This activation has been previously demonstrated in many teleosts after hypoxia, such as zebrafish [63,65], grass carp (*Ctenopharyngodon idellus*) [66], the European sea bass (*Dicentrarchus labrax*) [67] and the channel catfish [68]. *hif-2a* displayed similar expression patterns with *hif-1a.* This result was likely caused by their structurally analogous in the DNA-binding and dimerization domains [69], although the expression variation magnitude of *hif-2a* (< 5-fold) in the gill of the hybrid sturgeon was far less than that of *hif-1α* (>25-fold). Previous studies demonstrated that *hif-2α* is closely connected with the stimulation of matrix erythropoietin gene expression and maturation of the vascular network in response to hypoxia stress [70]. The expression levels of *hif-3a* in the MH and SH groups increased about 7- and 21-fold at 6 h after hypoxia, respectively, and returned to normal levels at 24 h. This observation was consistent with a previous study in zebrafish in response to acute hypoxia stress, which revealed that hypoxia stress induced *hif-3a* expression at both mRNA and protein levels [71]. However, the specific roles of *hif-3a* are not yet clear in fishes.

Several studies demonstrated that the expression of *hif-β* genes was constitutively expressed in an oxygen-independent way in higher vertebrates [28]. However, the expression levels of *hif-1β* and *hif-2β* in the hybrid sturgeon were significantly affected by hypoxia. So far, there is a paucity of data concerning the molecular response of the *hif-β* genes to hypoxia in sturgeon, while a previous observation regarding the gill of shrimp (*Litopenaeus vannamei*) [72] was consistent with the results of the *hif-1β* gene expression pattern in our study. Although there exist few studies about the *hif-2β* gene in aquatic animals in response to hypoxia, the strongly induced expression of the *hif-2β* gene in our study indicates that it might play certain roles in hypoxia adaptation in the hybrid sturgeon. It is worth noting that the expression of *hif-3β* in gill tissues was unaffected. At present, little is known about *hif-3β*. Its role in dealing with hypoxia remains to be further explored.

As known, the HIF pathway plays an integral role in the adaptive response to the hypoxia challenge in organisms [35,73]. Under the normoxia condition, *hif-a* usually undergoes hydroxylation and ubiquitination, followed by rapid proteasomal degradation [74]. The hydroxylation and ubiquitination processes were facilitated by the protein products of the *phd2* and *vhl* genes, respectively. By contrast, the *hif-a* degradation is rapidly inhibited, and the HIF pathway is activated under a hypoxia environment. In our results, the expression of the *vhl* gene was repressed by hypoxia in the liver tissues of the hybrid sturgeon, which is consistent with the activated HIF pathway. However, the increase expression was observed in the *phd2* gene under hypoxia. Interestingly, the opposite trends of *phd2* expression and DO levels were also detected in the liver tissues of the Wuchang bream (*Megalobrama amblycephala*) [75]. The mechanism of action is unclear, needing further study. Additionally, it was obvious that five of six *hif* genes, except for *hif-3a*, displayed increased expressions at 24 h after hypoxia challenge, suggesting the activation of the HIF pathway. The expression of some HIF-dependent genes was also increased at 24 h after hypoxia, including *glut1* (glucose transport), *gys* (glycogen synthesis), *ldha*, *gatm*, *angptl3* and *vegf*. The results mean that some other metabolic processes were dynamically regulated by the activated HIF pathway in the liver of the hybrid sturgeon, such as energy metabolism, oxidative stress response and angiogenesis [73,76]. At present, the research on the *hif* genes in the liver mainly focuses on *hif-1α* and *hif-2α*, whose main functions are regulating the development of fatty liver, liver fibrosis and liver cancer [77,78,79]. There are few studies on *hif-3*, and its role and function in gills and liver need to be further studied.

## 5. Conclusions

The present study suggests that both the RBC counts and the HGB concentration were observably increased after hypoxia stress. Serum cortisol concentrations were positively related to the severity of hypoxia stress. Moreover, several blood biochemical parameters, including AST, AKP, HBDB, LDH, GLU, TP and T-Bil, changed significantly after severe hypoxia. A complete set of six *hif* genes were characterized in hybrid sturgeon for the first time. Phylogenetic analysis provided additional evidence for their annotations. In addition, phylogenetic analysis showed that the hybrid sturgeon has a closer evolutionary distance to mammals compared to other fish species. The expression levels of *hif-1α*, *hif-2α*, *hif-3α*, *hif-1β* and *hif-2β* in the gill of the hybrid sturgeon were dramatically induced by hypoxia challenge, especially *hif-1α* and *hif-3α,* with more than 20-fold changes. Additionally, the meta-analysis results indicated that the HIF pathway was activated in the liver of the hybrid sturgeon at 24 after hypoxia. Overall, these results suggest that hypoxia, particularly severe hypoxia (1.00 ± 0.1 mg/L), could cause considerable stress for the hybrid sturgeon. Our results provide valuable information for the aquaculture industry to design effective strategies to avoid great economic losses caused by hypoxia.

## Figures and Tables

**Figure 1 genes-15-00743-f001:**
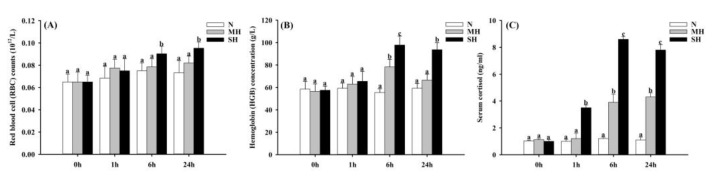
Effect of hypoxia on (**A**) red blood cell (RBC) counts, (**B**) hemoglobin (HGB) concentrations and (**C**) cortisol concentrations in hybrid sturgeons. Different letters represented significant differences (*p* < 0.05).

**Figure 2 genes-15-00743-f002:**
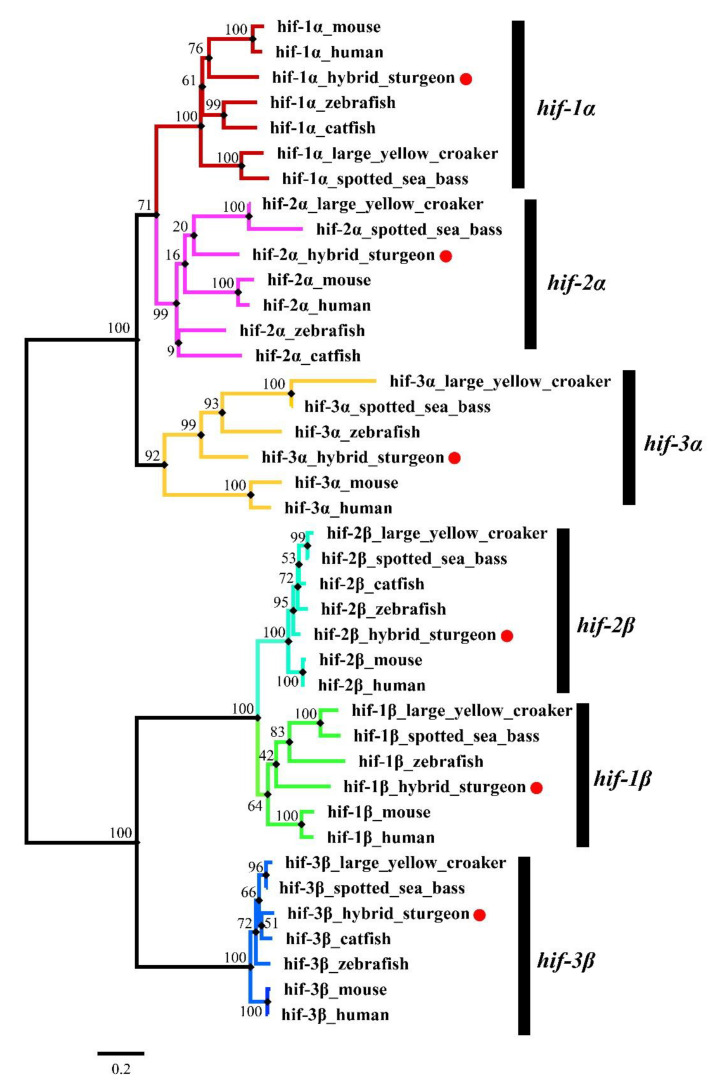
Phylogenetic relationships of *hif* genes in hybrid sturgeon and selected species. Multiple sequence alignments of *hif* amino acid sequences were performed using MUSCLE, and this phylogenetic tree was constructed using MEGA7 by the neighbor-joining (NJ) method and the Jones–Taylor–Thornton (JTT) model with 1000 bootstrap replicates. Different *hif* genes were classified into their corresponding clades with different colors, and the *hif* genes of hybrid sturgeon were labeled with red dots.

**Figure 3 genes-15-00743-f003:**
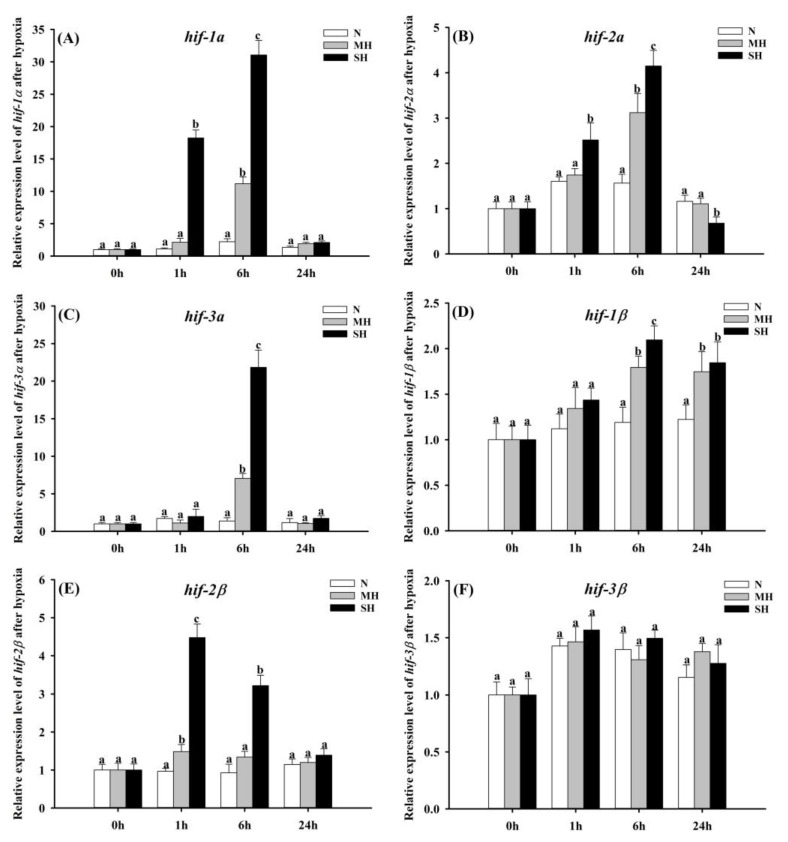
Expression patterns of *hif-1α* (**A**), *hif-2α* (**B**), *hif-3α* (**C**), *hif-1β* (**D**), *hif-2β* (**E**) and *hif-3β* (**F**) in gill tissues of hybrid sturgeon after hypoxia stress. Different letters represented significant differences (*p* < 0.05).

**Figure 4 genes-15-00743-f004:**
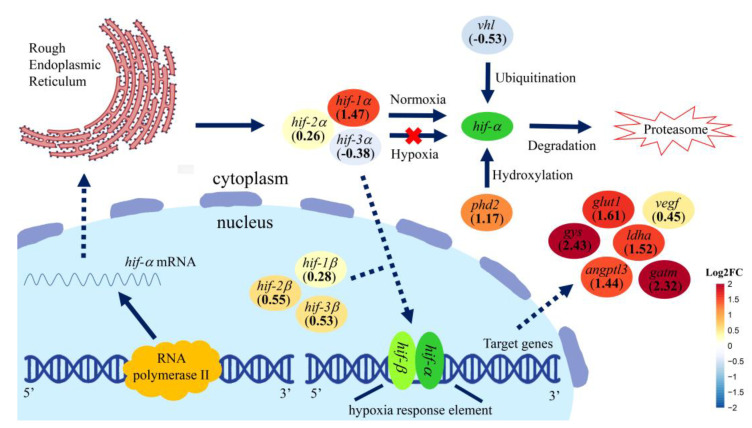
Schematic diagram of HIF pathway in liver of hybrid sturgeon 24 h after hypoxia challenge. The expressions of HIF pathway−related genes were determined by meta−analysis of RNA−Seq datasets. The specific expression levels of these function genes were determined in the SH group (n = 2) relative to N group (n = 2) based on the mapped reads using DESeq2 R package and shown in Appendix A. They are shown with continuous colors, from blue to red.

**Table 1 genes-15-00743-t001:** Serum biochemical parameters of hybrid sturgeon 24 h after hypoxia challenge.

Parameters		DO (mg/L)		
		N (DO:7.0)	MH (DO:3.5)	SH (DO:1.0)
Serum enzyme	ALT (U/L)	4.93 ± 0.80	5.37 ± 0.95	5.77 ± 0.86
	AST (U/L)	274.30 ± 45.39 ^a^	319.03 ± 24.78 ^ab^	387.83 ± 48.18 ^b^
	AKP (U/L)	275.20 ± 40.85 ^a^	230.93 ± 25.50 ^a^	174.37 ± 35.99 ^b^
	HBDH (U/L)	525.86 ± 74.16 ^a^	732.33 ± 159.33 ^ab^	868.06 ± 158.10 ^b^
	LDH (U/L)	950.73 ± 118.24 ^a^	1017.96 ± 127.37 ^a^	1589.96 ± 263.98 ^b^
Serum metabolites	GLU (mmol/L)	2.25 ± 0.33 ^a^	2.50 ± 0.24 ^a^	3.25 ± 0.13 ^b^
	TG (mmol/L)	6.62 ± 0.98	4.93 ± 0.55	5.34 ± 0.74
	TC (mmol/L)	2.22 ± 0.09	1.59 ± 0.34	2.04 ± 0.26
	TP (g/L)	14.63 ± 0.70 ^a^	16.10 ± 1.90 ^a^	20.70 ± 1.75 ^b^
	ALB (g/L)	4.20 ± 0.49	3.37 ± 0.60	4.33 ± 0.33
	T-Bil (umol/L)	12.00 ± 2.30 ^a^	14.63 ± 2.08 ^a^	19.26 ± 2.51 ^b^
	LA (mmol/L)	2.99 ± 0.11 ^a^	3.85 ± 0.09 ^b^	5.01 ± 0.14 ^c^
Serum inorganic ions	Ca (mmol/L)	2.01 ± 0.26	1.77 ± 0.22	1.88 ± 0.14
	Mg (mmol/L)	0.80 ± 0.09	0.83 ± 0.06	0.81 ± 0.09
	P (mmol/L)	2.70 ± 0.20	2.96 ± 0.20	2.83 ± 0.04

ALT: aminotransferase; AST: aspartate aminotransferase; AKP: alkaline phosphatase; HBDH: α-hydroxybutyrate dehydrogenase; LDH: lactate dehydrogenase; GLU: glucose; TP: total protein; ALB: albumin; TC: total cholesterol; TG: triacylglycerol; T-Bil: total bilirubin; LA: lactic acid; Ca: calcium; Mg: magnesium; P: phosphorus. Different letters represent significant differences (*p* < 0.05).

**Table 2 genes-15-00743-t002:** Characterizations of *hif* genes in hybrid sturgeon.

Group	Gene Name	CDS Size (bp)	Predicted Protein Length (aa)	Domain	GenBank Access Number
*hif-* *α*	*hif-1* *α*	2355	784	HLH, PAS, PAC, HIF-1, HIF-1a_CTAD	MN914157
	*hif-2* *α*	2562	854	HLH, PAS, PAC, HIF-1, HIF-1a_CTAD	MN914156
	*hif-3* *α*	1590	530	HLH, PAS, PAC, HIF-1	MN914155
*hif-* *β*	*hif-1* *β*	2343	781	HLH, PAS, PAC	MN914152
	*hif-2* *β*	1923	641	HLH, PAS, PAC	MN914153
	*hif-3* *β*	1890	630	HLH, PAS, PAC	MN914154

hif: hypoxia-inducible factor.

## Data Availability

The original contributions presented in this study are included in the article/Appendix A, further inquiries can be directed to the corresponding author.

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
