# Peer review of "Effects of Environmental Hypoxia on Serum Hematological and Biochemical Parameters, Hypoxia-Inducible Factor (hif) Gene Expression and HIF Pathway in Hybrid Sturgeon (Acipenser schrenckii ♂ × Acipenser baerii ♀)"

_genes, 2024, doi:10.3390/genes15060743_

Round 1
Reviewer 1 Report
Comments and Suggestions for Authors
Dear Editors,
Dear Authors,
In the reviewed manuscript the authors conducted a study to investigate the effects of environmental hypoxia on serum biochemical, hematological parameters, and the expression of hypoxia-inducible factor (HIF) genes in hybrid sturgeon (Acipenser Schrenckii ♂ × Acipenser Baerii ♀). The presented findings suggest that hypoxia triggers physiological responses in sturgeon, including increased RBC production and cortisol secretion, which are indicative of stress and adaptation mechanisms. Moreover, the recorded alterations in serum biochemical parameters indicate metabolic adjustments and potential tissue damage in response to hypoxia in the examined hybrid, reflecting the sturgeon's physiological adaptation to low oxygen conditions. The upregulation of HIF genes indicates their role in oxygen sensing and adaptive responses to hypoxia, potentially regulating gene expression involved in metabolic and physiological adjustments. The meta-analysis provides insights into the transcriptional regulation of genes involved in the HIF pathway, highlighting their role in metabolic modulating and angiogenesis promotion roles during hypoxia.
Overall, the study demonstrates the complex physiological and molecular responses of hybrid sturgeon to environmental hypoxia, shedding light on their adaptive mechanisms and potential biomarkers for hypoxia tolerance in aquaculture and conservation efforts.
However, the manuscript content require some improvements. Especially, information supplementation in the introduction chapter would greatly benefit the overall manuscript value and readability. Revision on language clarity improvements and grammar improvements is also reqiored.
Specific chapter by chapter review, including remarks, questions and suggested improvements are placed in the attached file.
Best regards,

Dear Editors,
Dear Authors,
In the reviewed manuscript the authors conducted a study to investigate the effects of environmental hypoxia on serum biochemical, hematological parameters, and the expression of hypoxia-inducible factor (HIF) genes in hybrid sturgeon (Acipenser Schrenckii ♂ × Acipenser Baerii ♀). The presented findings suggest that hypoxia triggers physiological responses in sturgeon, including increased RBC production and cortisol secretion, which are indicative of stress and adaptation mechanisms. Moreover, the recorded alterations in serum biochemical parameters indicate metabolic adjustments and potential tissue damage in response to hypoxia in the examined hybrid, reflecting the sturgeon's physiological adaptation to low oxygen conditions. The upregulation of HIF genes indicates their role in oxygen sensing and adaptive responses to hypoxia, potentially regulating gene expression involved in metabolic and physiological adjustments. The meta-analysis provides insights into the transcriptional regulation of genes involved in the HIF pathway, highlighting their role in metabolic modulating and angiogenesis promotion roles during hypoxia.
Overall, the study demonstrates the complex physiological and molecular responses of hybrid sturgeon to environmental hypoxia, shedding light on their adaptive mechanisms and potential biomarkers for hypoxia tolerance in aquaculture and conservation efforts.
However, the manuscript content require some improvements. Especially, information supplementation in the introduction chapter would greatly benefit the overall manuscript value and readability. Revision on language clarity improvements and grammar improvements is also reqiored.
Specific chapter by chapter review, including remarks, questions and suggested improvements are placed in the attached file.
Best regards,
Author Response
Dear editor and reviewers,
Thank you very much for the insightful comments on our manuscript . Those comments are valuable and very helpful for revising and improving our paper. We have studied these comments carefully and revised the manuscript according to the comments. The point-by-point response to the reviewer’s comments is uploaded in the attachment.

Reviewer 2 Report
Comments and Suggestions for Authors
1. Please authors follow the report entitled "Targeting the Mild-Hypoxia Driving Force for Metabolic and Muscle Transcriptional Reprogramming of Gilthead Sea Bream (Sparus aurata) Juveniles. Biology 2021, 10(5), 416".
2. 'The gene expression of HIF pathway-related were determined by meta-analysis of on-going RNA-Seq datasets", but in the section of Materials and Methods, the related papers on meta-analysisare not cited, and in the section of Results (Fig.4), could the authors provide numerical data on the schematic diagram of HIF pathway in liver of hybrid sturgeon at 24h after hypoxia challenge.
3. "The specific expression levels of these function genes were calculated as the FPKM values of SH group relative to N group and showed with continuous colors from blue to red", could the authors indicate the sample size (n) and the mean +/ SD of the FPKM values ?
4. Is there any data concerning about the expression levels of HIF pathway in gill of the fish at 24h after hypoxia challenge by meta-analysis?
5. Why the expression patterns of 6 hif genes in liver after hypoxia stress were missing in the report?
6. "After hypoxia, hif-1α and hif-3α with more than 20-fold changes in the gills", but the trend seemed the same in the liver by the meta-analysis, could the authors discuss the physiological roles of various hif isoforms in the liver, although "it was obvious that 5 of 6 hif genes, except for hif-3a, displayed increased expressions at 24 h after hypoxia challenge", could the authors explain the different roles of hif-3a in gills or liver?
7. To exert meta-analysis, is there any information about the variation on the basal levels among different individuals ?
Round 2
Reviewer 1 Report
Comments and Suggestions for Authors
Dear Editors,
Dear Authors,
The manuscript entitled: “Effects of environmental hypoxia on serum biochemical, haematological parameters, hypoxia-inducible factor (hif) genes expression and HIF pathway in the hybrid sturgeon (Acipenser Schrenckii ♂ × Acipenser Baerii ♀)” has been significantly improved. The Authors have regarded all my remarks and suggestions.
Thank you for all answers!
Best regards,
Comments on the Quality of English LanguageDear Editors,
Dear Authors,
The manuscript entitled: “Effects of environmental hypoxia on serum biochemical, haematological parameters, hypoxia-inducible factor (hif) genes expression and HIF pathway in the hybrid sturgeon (Acipenser Schrenckii ♂ × Acipenser Baerii ♀)” has been significantly improved. The Authors have regarded all my remarks and suggestions.
Thank you for all answers!
Best regards,
Author Response
Thank you for your recognition, and thanks again for your valuable comments.
Reviewer 2 Report
Comments and Suggestions for Authors
As replied by the authors:
"The bioinformatic pipelines have been constructed in the previous studies of zebrafish [38], gilthead sea bream (Sparus aurata) [39] and flounder (Paralichthys olivaceus) [40]. The high-quality reads were mapped to the above-mentioned assembly transcripts using Hisat2 v2.2.1 software [41]"
Question: Fish is polyploid, How the authors do deal with the isoforms of given genes in the Fig. 4 as well as Table S2 as the hifs ?
"The specific expression levels of these function genes were determined in the SH group (n=2) relative to N group (n=2) based on the mapped reads using DESeq2 R package", and "Two SH and N samples were performed for the calculation of expression levels. As defined, the DESeq2 R package cannot generate any values of mean +/ SD", as well as "The specific expression levels of these function genes were determined in the SH group relative to N group based on the mapped reads using DESeq2 R package"
Question: why the sample size in each group is two? and how the authors can confirm that the individual variation is not significant? although the data were mixed or melanged by the R-package.
"Unfortunately, transcriptome analysis of gills was not performed because of insufficient samples"
Question: I can tell, thanks for the reply.
Author Response
"The bioinformatic pipelines have been constructed in the previous studies of zebrafish [38], gilthead sea bream (Sparus aurata) [39] and flounder (Paralichthys olivaceus) [40]. The high-quality reads were mapped to the above-mentioned assembly transcripts using Hisat2 v2.2.1 software [41]"
Question: Fish is polyploid, How the authors do deal with the isoforms of given genes in the Fig. 4 as well as Table S2 as the hifs ?
Response: Thanks for your valuable comments. The reference genome of hybrid sturgeon (Acipenser Schrenckii ♂ × Acipenser Baerii ♀) has not been published until now. Therefore, RNA-seq datasets were selected for the de novo assembly using the TRINITY software to obtain the transcript sequences. Isoforms of hif genes in zebrafish were downloaded from the NCBI database, working as queries to search their homologues genes in the hybrid sturgeon. Based on the best-hit results, the isoforms of hif genes in hybrid sturgeon were identified and characterized in the present study.
"The specific expression levels of these function genes were determined in the SH group (n=2) relative to N group (n=2) based on the mapped reads using DESeq2 R package", and "Two SH and N samples were performed for the calculation of expression levels. As defined, the DESeq2 R package cannot generate any values of mean +/ SD", as well as "The specific expression levels of these function genes were determined in the SH group relative to N group based on the mapped reads using DESeq2 R package"
Question: why the sample size in each group is two? and how the authors can confirm that the individual variation is not significant? although the data were mixed or melanged by the R-package.
Response: Thanks for your excellent comment. In the present study, these RNA-seq datasets were generated several years ago. Because of the limitation of sequencing costing, only 2 samples were conducted for the control and treatment groups. However, to reduce the variations of biological samples, equal-quality liver tissues from three individuals were pooled together and constructed for the RNA-seq library. The significance of gene expression was determined by the P-value, calculated by the DESeq2 R package.